# Acute Biliary Pancreatitis Management during the Coronavirus Disease 2019 Pandemic

**DOI:** 10.3390/healthcare10071284

**Published:** 2022-07-11

**Authors:** Elif Çolak, Ahmet Burak Çiftci

**Affiliations:** Department of General Surgery, Samsun University, Samsun Training and Research Hospital, Samsun 55090, Turkey; elif.colak@samsun.edu.tr

**Keywords:** acute biliary pancreatitis, cholecystectomy, coronavirus disease 2019, severe acute respiratory syndrome coronavirus-2, quick sequential organ failure assessment, World Society of Emergency Surgery sepsis severity score

## Abstract

(1) Objective: We aimed to analyze and describe the management of acute biliary pancreatitis (ABP) during the coronavirus disease 2019 (COVID-19) pandemic. (2) Methods: This was a retrospective cohort study among patients with ABP during a control period (16 March 2019 to 15 March 2020; period 1) and a COVID-19 period (16 March 2020 to 15 March 2021; period 2). (3) Results: We included 89 patients with ABP, being 58 in period 1 and 31 in period 2. The mean patient age was 62.75 ± 16.59 years, and 51 (57.3%) patients were women. The Quick Sequential Organ Failure Assessment score for sepsis and World Society of Emergency Surgery Sepsis Severity Score were significantly higher among patients in period 2. Twenty-two patients (37.9%) in period 1 and six (19.3%) in period 2 underwent cholecystectomy. There were no significant differences in surgical interventions between the two periods. The hospital mortality rate was 3.4 and 19.3% in period 1 and period 2, respectively. Mortality was significantly higher in period 2. Conclusion: During the COVID-19 pandemic, we observed a significant reduction in the number of patients with ABP but increased severity and mortality. Multicenter studies with more patients are needed to obtain additional evidence regarding ABP management during the COVID-19 pandemic.

## 1. Introduction

The outbreak of coronavirus disease 2019 (COVID-19) was declared a global pandemic by the World Health Organization on 11 March 2020 [1]. Since then, unexpected changes have taken place in health care systems worldwide. In many countries, medical care has shifted according to the burden of COVID-19. After these changes, a sharp decrease in non-COVID-19 patient admissions to the emergency department has been observed [2,3,4,5]. Multiple studies have shown a considerable reduction in emergency surgical admissions, ranging from 14 to 86% [6,7,8,9]. What happened to emergency patients during the pandemic and what are the impacts of late or non-presentation to the hospital on outcomes among emergency patients, are critical questions.

It is expected that during a pandemic, late presentation, or failure to present to the emergency department would lead to more complicated diseases in some patients. Studies have shown an increase in cases of complicated and perforated appendicitis during the COVID-19 pandemic [10,11]. Additionally, palliative interventions other than early surgery have been recommended for patients with acute cholecystitis during this period [12,13]. However, no studies have investigated the effect of the pandemic on outcomes in patients with acute biliary pancreatitis (ABP).

Acute pancreatitis is an inflammatory disease of the pancreas most often caused by gallstones or alcohol consumption. Apart from these etiological causes, acute pancreatitis associated with many different viruses has also been reported in the literature [14]. Although the relationship between SARS CoV-2 and acute pancreatitis is not certain, it has been suggested in some case reports in the literature that COVID-19 may cause pancreatitis with direct damage to the pancreas [15]. In most patients, the disease has a mild course, and moderate fluid resuscitation and pain management can result in rapid clinical improvement. By comparison, the severe form of pancreatitis, comprising approximately 20 to 30% of patients, is a life-threatening disease with hospital mortality rates of approximately 15% [16]. Initial assessment of the severity of acute pancreatitis is vital in determining further medical treatment. Intravenous fluid replacement is essential to treat fluid loss owing to third-space shifts, vomiting, and increased vascular permeability caused by inflammatory mediators. Hemodynamic status should be assessed immediately upon presentation and resuscitative measures begun as needed. Patients with organ failure should be admitted to an intensive care unit (ICU). Hydration should be provided to all patients unless cardiovascular or renal comorbidities preclude this. Early intravenous hydration is most beneficial within the first 12–24 h, after which time it may have a little benefit [17,18,19].

In light of these factors, late admission of patients with acute pancreatitis and inadequate initial treatment may increase the severity of the disease. Thus, we aimed to investigate how a sudden disruption in seeking surgical care during the COVID-19 pandemic might affect severity rates and outcomes in ABP. The primary aim of this study was to identify the impact of the COVID-19 pandemic on rates of admission and severity of ABP. The secondary aim was to compare the management of ABP before and during the COVID-19 pandemic.

## 2. Materials and Methods

### 2.1. Study Design

This retrospective study was conducted at the University of Samsun, Samsun Training and Research Hospital in Samsun, Turkey. We reviewed the hospital records for all consecutive adult patients with ABP during two discrete periods: a control period from 16 March 2019 to 15 March 2020 (period 1) and a COVID-19 pandemic period between 16 March 2020 and 15 March 2021 (period 2). The first COVID-19 case in our region was reported on 16 March 2020. The COVID-19 period was defined based on the first confirmed case in the study area.

The inclusion criteria were patients over the age of 18 years who were admitted to the hospital for ABP, diagnosed in the emergency department, or during hospitalization via an imaging test (ultrasound, computerized tomography (CT)). Exclusion criteria were patients under the age of 18 years, those who had acute pancreatitis without a biliary etiology, and patients presenting with chronic pancreatitis, pancreatic malignancy, or pregnancy. Pregnant women were excluded from the study because there is no obstetrics and gynecology clinic in our hospital. Non-biliary acute pancreatitis were not included in this study because these patients are followed by the gastroenterology department in our hospital and are not consulted to general surgery.

Pancreatitis was confirmed if a patient had at least two of the following: characteristic pain, lipase levels more than three times the upper limit of standard, or radiological evidence of pancreatitis. Confirmation of a biliary etiology was performed via an imaging test (ultrasound, CT).

Disease severity was assessed using the Revised Atlanta Classification (RAC) [20]. This classification identifies two phases, early and late, and severity is classified as either mild, moderate, or severe. The mild form (interstitial edematous pancreatitis) involves no organ failure (OF), no local or systemic complications, and usually resolves in the first week. If there is transient (less than 48 h) OF, local complications, or exacerbation of the comorbid disease, severity is classified as moderate. Patients with persistent (more than 48 h) OF have a severe form of the disease. OF is defined as cardiac failure (systolic blood pressure (SBP) <90 mmHg), pulmonary insufficiency (partial pressure of oxygen in arterial blood <60 mmHg on room air or mechanical ventilation requirement), and renal failure (serum creatinine level >2 mg/dL or need for hemodialysis). Persistent OF is defined as OF lasting for more than 48 h.

Classical Balthazar scoring system was used for radiological scoring [21]. In this classification, there are 5 different categories, from A to E. Grade A indicates a normal pancreas, grade B indicates pancreatic enlargement, grade C indicates peripancreatic fat present in inflammatory changes, grade D indicates single peripancreatic fluid collection, and grade E indicates 2 or more fluid collections or gas bubbles in, or adjacent to, the pancreas.

The severity classification based on the World Society of Emergency Surgery Sepsis Severity Score (WSES SSS) is as follows: 3, low severity; 4–6, intermediate severity; and 7, high severity. The quick Sequential Organ Failure Assessment (qSOFA) score includes three criteria, assigning one point each for low blood pressure (SBP < 100 mmHg), high respiratory rate (>22 breaths per minute), or altered mentation (Glasgow Coma Scale < 15).

All acute surgical patients waiting for hospital admission and urgent surgery were screened for COVID-19 infection in the emergency department of our hospital during the COVID-19 pandemic period.

### 2.2. Data Analysis and Evaluation

The following information was collected from the patient records during the first episode of ABP on admission: age, sex, body mass index (BMI), symptom duration, comorbidities, Charlson Comorbidity Index (CCI), vital signs (body temperature, pulse rate, and SBP), hematology findings (white blood cell count, neutrophil count, platelets, C-reactive protein, amylase, lipase, creatinine, aspartate aminotransferase, alanine aminotransferase, lactate, and total and conjugated bilirubin), Balthazar CT score, qSOFA score for sepsis, Bedside Index of Severity in Acute Pancreatitis score (BISAP), Glasgow–Imrie Criteria for Severity of Acute Pancreatitis, Ranson criteria for pancreatitis, and WSES SSS. Additionally, we recorded the rates of magnetic resonance cholangiopancreatography (MRCP), endoscopic retrograde cholangiopancreatography (ERCP), percutaneous CT-guided fine-needle aspiration biopsy (CT-guided FNAB), cholecystectomy, ICU admission, and hospital mortality, as well as the type of cholecystectomy and length of hospital stay.

Our study included the following therapeutic interventions: endoscopic sphincterotomy, endoscopic intervention, and surgical intervention. Endoscopic interventions included endoscopic drainage and necrosectomy for infected acute necrotic collection or walled-off necrosis (WON). Surgical interventions were defined as percutaneous drainage, open/laparoscopic debridement, and drainage with or without pancreatic resection. The diagnosis of WON was conducted according to the 2012 RAC and referred to necrotic tissue contained within an enhancing wall of reactive tissue that occurred ≥4 weeks after the onset of ABP.18

### 2.3. Outcome Measures

The primary outcomes of interest were differences in the rate of ABP admission and ABP severity. Secondary outcomes of interest were therapeutic interventions, admission to the ICU, duration of ICU stay, length of hospital stay, and hospital mortality. The severity of ABP was assessed based on the RAC. BISAP, qSOFA, Glasgow—Imrie Criteria for Severity of Acute Pancreatitis, Ranson criteria for pancreatitis, and WSES SSS.

### 2.4. Statistical Analysis

IBM SPSS software version 20 was used for the statistical analysis (IBM Corp., Armonk, NY, USA). Scale data were tested for normality using the Shapiro–Wilk test. Non-parametric data were tested with the Mann–Whitney test. Parametrically distributed data were tested using the Student *t*-test. The chi-square test was used to compare differences in categorical variables, and Fisher’s exact test was used for small sample sizes (expected frequency of the test variable less than 5). A *p*-value < 0.05 was considered to indicate statistical significance.

### 2.5. Ethical Considerations

The study was approved by the Institutional Review Board of Samsun Training and Research Hospital (GOKA/2021/9/8). We confirm that all methods were carried out under relevant guidelines and regulations. To avoid the potential risk of bias, one researcher extracted all the data, and the second researcher independently checked the data extraction forms for accuracy and detail. Patient consent was not required owing to the retrospective nature of the study. This study followed the guidelines outlined in the Strengthening the Reporting of Observational Studies in Epidemiology (STROBE) Statement [22].

## 3. Results

### 3.1. Patient Characteristics on Admission

During the study period, 89 patients with ABP were identified, with 58 in period 1, and 31 in period 2. The total number of patients hospitalized after emergency room admission in period 1 was 324, and 274 in period 2. Among the reasons for hospitalization, acute biliary pancreatitis diagnosis rates were 17.9 and 11.3%, respectively. This difference was statistically significant (*p* = 0.024). The mean age of patients was 62.75 ± 16.59 years, and 51 (57.3%) were women. Most (57%) of the study population was over 60 years of age. In total, 44.9% of patients had diabetes mellitus, 12.4% had hypertension, and 37% had no comorbid conditions. Among all patients, 40% had a BMI of 30–35 kg/m^2^. Basic demographics are outlined in Table 1. There were no significant differences in mean age, sex, BMI, or CCI between the two periods and no significant differences regarding hematological parameters, except lactate. Lactate levels were significantly higher in period 2 (*p* = 0.012).

### 3.2. Severity, Treatment, and Clinical Outcomes

The severity of index disease according to the RAC is outlined in Table 2. According to the RAC, most of the patients in both groups were assessed as mild. However, there was a significant difference in the severity of disease among patients with ABP between the two periods (*p* = 0.040).

Table 3 presents the results of comparisons between patients in the two periods in terms of mean qSOFA, BISAP, Glasgow–Imrie Criteria for Severity of Acute Pancreatitis, Ranson criteria for pancreatitis, Balthazar score, and WSES SSS. Scores for qSOFA, WSES SSS and Balthazar were significantly higher among patients in period 2 (respectively, *p* = 0.031, *p* = 0.032, and *p* = 0.004).

Fifty-four patients (93.1%) had a qSOFA score of 0, three (5.2%) patients had a qSOFA score of 1, and one (1.7%) patient had a score of 2 in period 1. No patients had a qSOFA score of 3 in period 1. Twenty-three patients (74.1%) had a qSOFA score of 0, five (16.1%) patients had a qSOFA score of 1, two (6.5) patients had a score of 2, and one (3.3%) patient had a score of 3, in period 2. Severity classification based on WSES SSS was as follows: 57 patients (98.2%) had low severity scores (≤3) and 1 patient (1.8%) had a high severity score in period 1. Twenty-five patients (80.6%) had a low severity score, four patients (12.9%) had an intermediate score, and two patients (6.5%) had a high severity score in period 2. Distribution of patients’ Balthazar grades according to periods are shown in Table 4. Diagnostic imaging and therapeutic interventions are shown in Table 5. There were no significant differences found between the two periods regarding diagnostic imaging and therapeutic interventions. Twenty-two patients (37.9%) in period 1, and six (19.3%) patients in period 2 underwent cholecystectomy. There was no significant difference between the two periods regarding index cholecystectomy. Cholecystectomy was performed laparoscopically in 18 (81.8%) patients during period 1, and in 5 (83.3%) patients during period 2. There was no difference found between the two periods regarding laparoscopic cholecystectomy.

### 3.3. Outcomes

Six patients (10.3%) in period 1, and ten (32.2%) patients in period 2 were admitted to the ICU. The rate of ICU admission was significantly higher in period 2 (*p* = 0.010). The median length of hospital stay was 5 (1–40) days in period 1, and 4 (2–75) days in period 2. There was no difference found between the two periods. According to tomography images, necrotizing pancreatitis developed in two patients (3.4%) in period 1, and in five patients (16.1%) in period 2. Two patients in period 2 were diagnosed with COVID-19 infection at admission; one patient was diagnosed with COVID-19 after admission (on day 7). Three patients were positive for COVID-19 infection, and all of these patients eventually died. The severity of ABP was significantly worse in SARS-CoV-2-positive patients, with 100% of patients in this group developing severe pancreatitis. The hospital mortality rate was 3.4 and 19.3% in period 1 and period 2, respectively. Mortality was significantly higher in period 2 (*p* = 0.012).

## 4. Discussion

ABP is the most common form of acute pancreatitis encountered by physicians in the western world [23]. This study showed that significantly fewer patients with ABP were admitted to our emergency department during the COVID-19 pandemic than during the pre-pandemic period. The findings of this study are similar to those of studies in the United Kingdom, Italy, and Spain, which reported significant reductions in general surgical admissions, ranging from 14 to 86% [2,3,6]. According to their analysis of data from 36 emergency departments, Slagman et al. also reported a significant decrease in medical emergencies among all disciplines after the introduction of contact restrictions during the COVID-19 pandemic [4].

The above findings could be related to the public measures implemented to reduce population mobility, hesitation in going to an emergency department owing to the risk of contagion, and spontaneous resolution of mild ABP self-limited with symptomatic treatment at home. Late admission to the emergency department may reduce the overall rate of ABP diagnosis and might increase the severity of ABP presentation.

We expected delays in patient presentations to the emergency department during the COVID-19 pandemic. However, when the time between symptom onset and emergency admission was compared, we found no significant difference between the groups (*p* = 0.186). From this point of view, it can be speculated that patients with acute biliary pancreatitis were not late in applying to the emergency department during the pandemic period.

As initial treatment is vital in managing ABP, aggressive intravenous fluid replacement should be administered to treat fluid loss caused by third-space shifts, vomiting, and increased vascular permeability. We found that lactate levels on admission were significantly higher in patients during period 2, which could be related to inadequate fluid replacement.

Prompt evaluation of the severity of ABP is essential to predict patient outcomes, estimate prognosis, and determine the need for ICU care. We found that qSOFA and WSES scores were higher among patients in period 2 than those in period 1. Clinical signs with higher qSOFA and WSES scores indicate a higher rate of a severe disease course, similar to other diseases in the general patient population. However, the main reason for the high qSOFA and WSESS scores during the pandemic period was thought to be due to the fact that these scoring systems were essentially used in sepsis, and the development of sepsis due to COVID-19 pneumonia in three patients during the pandemic period may have contributed to this difference. A situation similar to our findings has been reported in patients with acute appendicitis. Many studies have shown a decrease in admissions for acute appendicitis as well as an increase in the number of complicated appendicitis cases during the COVID-19 pandemic [10,11]. Willms et al. reported that in the New York metropolitan area, the overall number of patients with appendicitis decreased from 1027 in 2019 to 888 (−13.5%, *p* = 0.003), and the rate of complicated appendicitis rose to 64.4% (*p* = 0.012) during the COVID-19 lockdown [11].

Tomography has an important role in the diagnosis and follow-up of acute biliary pancreatitis. In this study, abdominal tomography was used very frequently in our patients, either in their admission to the emergency room or in their follow-up, if needed, and the images were classified with Balthazar grade. In this study, Balthazar grades were found to be significantly higher in period 2 (*p* = 0.004). It is a matter of debate why the tomography grades were higher during the pandemic period. Considering that ABP patients’ admission to the emergency department was not delayed during this period, it should be considered that other factors may have affected the severity of tomography. It is possible that SARS-CoV-2 virus may have been one of these factors, but the authors could not reach a definite conclusion on this subject according to the available data.

There are increasingly more studies that have connected acute pancreatitis with SARS-CoV-2 infection [24]. The expression of angiotensin-converting enzyme 2 (ACE2) in pancreatic cells renders the pancreas a potential target for SARS-CoV-2. In a study investigating the relationship between SARS-CoV-2 infection and acute biliary pancreatitis, Meriç et al. [25] compared the pandemic period with the pre-pandemic period to reach a conclusion. As a result of this research covering 6-month periods, they emphasized that clinicians dealing with acute pancreatitis should be aware of a possible relationship between the presence of COVID-19 and pancreatitis. They also found that acute biliary pancreatitis cases decreased during the pandemic period, but there was no difference in cholecystectomy rates, mortality and intensive care need between the two periods. Similarly, we found a decrease in acute biliary pancreatitis patients in the pandemic period and cholecystectomy rates were similar. However, unlike this study, we found a significant increase in the need for intensive care and mortality in patients during the pandemic period. The reason for this difference can be attributed to patients who applied to or were referred to our tertiary reference hospital arrived with more severe pancreatitis. Additionally, sepsis and mortality due to COVID-19 infection may have affected our results.

In our study, we found that three of six patients who died in period 2 had concurrent COVID-19 infection. All three of these patients had persistent OF and necrotizing pancreatitis. It is difficult to make a definitive interpretation of the cause of death in these cases because both diseases can be fatal. However, we can conclude that both COVID-19 and ABP can lead to severe illness and death.

In the present study, we focused on the management of ABP before and during the COVID-19 pandemic. Although the number of patients was small in our study, we performed cholecystectomy at a similar rate in both periods among patients with mild ABP. Cholecystectomy was performed laparoscopically in 18 patients during period 1, and in 5 patients during period 2. In fact, our cholecystectomy rates at the same hospitalization remained low, contrary to the recommendations of the guidelines on this subject. The reason for this may be the thought of some surgeons that dissection difficulties would occur after pancreatitis and patients were afraid of the high risk of operation. On the other hand, no changes to the surgical management of mild ABP were implemented in our clinic in response to the possible increased risk of exposure to COVID-19 in laparoscopic surgery. All these patients were discharged in good condition.

We also used advanced imaging modalities and interventional procedures such as MRCP, ERCP, CT-guided FNAB, percutaneous drainage, and endoscopic drainage at similar rates during both periods. This can be interpreted as indicating that the standard of emergency surgical care for patients with ABP was maintained in our hospital.

The main limitation of this study is its retrospective design and use of retrospective data obtained from electronic medical records; the interpretation of these data might include bias. Another limitation is its single center design. While the COVID-19 pandemic continues, a single-centered retrospective study was preferred in order to reach the data as quickly as possible and share it with the medical literature as fast as we can. Additionally, our findings may be influenced by inter-individual variability in clinical decision-making.

Despite the recommendations of many surgical societies for the management of ABP, there is no clear evidence regarding ABP management during the COVID-19 pandemic. To our knowledge, this is the first study to analyze the patient number, severity, and management of ABP during the COVID-19 pandemic. In this paper, we analyzed and described ABP management at our center during a period with high incidence during the COVID-19 pandemic to provide evidence that will be helpful in emergency surgical patient management during a public health emergency.

## 5. Conclusions

To conclude, we found a significant reduction in the number of patients with ABP who presented to our hospital during the COVID-19 pandemic. However, there was an increase in the severity of disease observed during this period. Analysis and evaluation of patients with ABP during the COVID-19 pandemic is important in developing strategies to confront future public health crises. Thus, multicenter studies with a larger number of patients are needed to obtain further evidence regarding ABP management during the COVID-19 pandemic.

## Figures and Tables

**Table 1 healthcare-10-01284-t001:** Patient demographics and baseline characteristics on admission.

	Period 1 (n *=* 58)	Period 2 (n = 31)	*p*-Value
^a^ Age (years)	61.1 ± 16.9	62.0 ± 16.0	0.767
^b^ Sex, male/female	27/31 (46.6%/53.4%)	11/20 (35.5%/64.5%)	0.372
^a^ BMI (kg/m^2^)	29.3 ± 4.2	30.7 ± 5.0	0.181
^b^ Symptom duration at admission			0.186
<48 h	45 (77.6%)	20 (64.5%)	
>48 h	13 (22.4%)	11 (35.5%)	
^c^ Charlson Comorbidity Index	3 (0–4.25)	3 (1–5)	0.569
**Physical and blood examinations**			
^c^ Body temperature (°C)	36.7 (36.4–37.0)	36.9 (36.2–37.4)	0.091
^c^ Pulse rate (/minute)	68 (66–74)	72 (66–78)	0.424
^c^ SBP (mmHg)	120 (110–140)	120 (110–140)	0.546
^c^ WBC (×10^3^/mL)	10.9 (8.6–14.0)	12.4 (9.1–13.9)	0.494
^c^ Neutrophil count (×10^3^/mL)	8.6 (5.8–11.6)	9.3 (6–12)	0.590
^c^ Platelets (×10^4^/mL)	22.85 (18.0–32.2)	25.2 (18.7–31.1)	0.217
^c^ CRP (mg/dL)	7.24 (1.4–11.8)	8.04 (2.4–12.6)	0.494
^c^ Amylase (U/L)	814 (471–1758)	1125 (416–2353)	0.552
^c^ Lipase (U/L)	1765 (725–5256)	2972 (777–7526)	0.438
^c^ AST (U/L)	154 (53–283)	106 (45–339)	0.667
^c^ ALT (U/L)	121 (40–249)	78 (24–257)	0.532
^c^ Total bilirubin (mg/dL)	1.3 (0.6–2.4)	1.4 (0.6–1.8)	0.645
^c^ Creatinine (mg/dL)	0.8 (0.7–1.1)	0.8 (0.6–1.1)	0.681
^c^ Lactate (mmol/L)	1.4 (1.2–1.8)	1.8 (1.4–2.5)	**0.012 ***

* *p* < 0.05. AST, aspartate aminotransferase; ALT, alanine aminotransferase; BMI, body mass index; CRP, C-reactive protein; SD, standard deviation; SBP, systolic blood pressure; WBC, white blood cell. ^a^ Student’s *t*-test with mean ± standard deviation (SD). ^b^ Chi-square with n (%). ^c^ Mann–Whitney U-test with median (interquartile range, IQR).

**Table 2 healthcare-10-01284-t002:** Revised Atlanta Classification of patients in period 1 and period 2.

	Period 1, n = 58	Period 2, n = 31	*p*-Value
Mild	49 (84.5%)	21 (67.7%)	0.04
Moderately severe	7 (12.0%)	4 (12.9%)	
Severe	2 (3.5%)	6 (19.3%)	

**Table 3 healthcare-10-01284-t003:** Comparison of prognostic scores between period 1 and period 2.

	Period 1, n = 58	Period 2, n = 31	*p*-Value
* qSOFA score	0 (0–0)	0 (0–1)	0.031
* WSESS SSS	0 (0–2)	2 (0–3)	0.032
BISAP Score	2 (0.75–2)	1 (1–2)	0.939
Glasgow Score	2 (1–2)	2 (1–3)	0.402
Ranson Score	2 (1–3)	3 (1–5)	0.053
^‡^ * Balthazar Score	2 (1–3)	3 (1–4)	0.004

* *p* < 0.05. BISAP; Bedside Index of Severity in Acute Pancreatitis, Glasgow; Glasgow–Imrie Criteria for Severity of Acute Pancreatitis, qSOFA; quick Sequential Organ Failure Assessment score for sepsis, Ranson; Ranson criteria for pancreatitis, WSES SSS; World Society of Emergency Surgery Sepsis Severity Score. Data are presented as median (interquartile range, IQR). ^‡^ We translated Balthazar score A to E into a point scale 1 to 5.

**Table 4 healthcare-10-01284-t004:** Distribution of patients’ Balthazar grades according to periods.

Grade	Period 1, n = 58	Period 2, n = 31	Total, n = 89	*p*-Value
A	17 (29.3%)	9 (29.0%)	26 (29.2%)	
B	20 (34.5%)	5 (16.1%)	25 (28.1%)	
C	15 (25.9%)	6 (19.4%)	21 (23.6%)	0.046
D	4 (6.9%)	6 (19.4%)	10 (11.2%)	
E	2 (3.4%)	5 (16.1%)	7 (7.9%)	

**Table 5 healthcare-10-01284-t005:** Comparison of diagnostic imaging and therapeutic interventions between period 1 and period 2.

	Period 1, n = 58	Period 2, n = 31	*p*-Value
CT	51 (87.9%)	30 (96.7%)	0.165
MRCP	13 (22.4%)	10 (32.2%)	0.312
ERCP	4 (6.8%)	2 (6.4%)	0.654
PD	2 (3.4%)	3 (9.6%)	0.337
CT-guided FNAB	2 (3.4%)	3 (9.6%)	0.337
ED	1 (1.7%)	1 (3.2%)	0.578

CT, computed tomography; CT-guided FNAB, percutaneous CT-guided fine-needle aspiration biopsy; ED, endoscopic drainage; ERCP, endoscopic retrograde cholangiopancreatography; MRCP, magnetic resonance cholangiopancreatography; PD, percutaneous drainage.

## Data Availability

The datasets generated in the current study are available from the corresponding author on reasonable request.

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
