# Peer review of "Acute Biliary Pancreatitis Management during the Coronavirus Disease 2019 Pandemic"

_healthcare, 2022, doi:10.3390/healthcare10071284_

Round 1
Reviewer 1 Report
The main concern regarding that paper is a retrospective nature of the study and small study group. Although authors are aware of these limitations (line 254-257, 270-272), it is not explained why they decided to perform retrospective single-centre study.
Introduction should be improved. Please include information about an association between acute pancreatitis and SARS-CoV-2 infection. The authors should also highlight why they decided to analyse only biliary pancreatitis instead of acute pancreatitis regardless of aetiology.
Please rationale your study since recently there has been published a similar paper "COVID-19 and acute biliary pancreatitis: comparative analysis between the normal period and COVID-19 pandemic" (MEris S. et al. Ann Ital Chir. 2021;92:728-731. PMID: 35166231). Please discuss results of that study with your findings.
Important missing data which should be analysed are: diagnostic delay - time from symptoms onset to hospital admission and results of imaging studies.
Since the standard management of biliary acute pancreatitis is cholecystectomy during the same hospitalization after acute symptoms have subsided, please explain sucha low rate of choldecystectomy in your group of patients.
Data from the text are repeated in Tables - please improve that.
Reviewer 2 Report
Authors have evaluated the hospitalizations due to ABP during COVID pandemic and compared to previous year.
Methods:
1. It will be useful to see hospitalization rates due to ABP rather than absolute number. Meaning how many ABP per 1000 or 10000 ER visits. Just demonstrating absolute value carries little significance as the hospitalization rates dropped world wide.
2. Can the authors look at annual admission of ABP over 2 to 3 years prior to COVID pandemic. Just comparing 1 year prior to COVID is kind of a snapshot and looking through 2 to 4 prior years would be ideal.
Results:
Any hemorrhagic pancreatitis. ? How many had necrotizing pancreatitis?
How many had sepsis or severe sepsis? Because only qSOFA and WSESS scores which are sepsis specific is different however AP specific scores did not show any significant difference. What is the reason? Did they have systemic sepsis/septicemia which also involved pancreas and there by having worsened outcomes?
Discussion: Add reference to statement in line 204 and 205, Page 6. I am not sure if we can say significantly low number of ABP especially when we don't have a denominator.
